# Safety outcomes for incident responders operating on high speed roads: An analysis of the relationship with behaviour, motivation and role clarity

Sharon Newnam[1]*, Amanda Stephens[1], Carlyn Muir[1], Simon Bruce[2], Tim Austin[3], Tony Mazzeo[4]

1 Monash University Accident Research Centre, Monash University, Clayton, VIC, Australia, 2 Holmesglen Institute, Melbourne, Victoria, Australia, 3 Visual Learning Design, Victoria, Australia, 4 Broadspectrum, Transport ANZ, Victoria, Australia

* sharon.newnam@monash.edu

**Data Availability Statement:** All relevant data are within the paper and its Supporting Information files.

## Abstract

High-speed roads present a considerable level of risk for frontline workers operating in these environments. To optimise safety, prevention activities need to target the key skills required to mitigate risk. The aim of this research was to explore the behavioural (compliance, participation, voice), motivational (safety motivation) and work demand (role clarity) factors that influence safety outcomes for incident responders working on high-speed roads. Safety outcomes included secondary incidents and near misses with passing vehicles. A total of 295 complete survey responses were received from six emergency service and incident response agencies in one Australian state. Data were analysed using structural equation modelling. The results showed that higher levels of safety voice, safety motivation and, role clarity were significantly associated with safer self-reported safety outcomes after controlling for the number of incidents attended. The findings from this study will be used to guide the development of a training program to improve the cognitive, behavioural and perceptual skills of incident responders operating on high-speed roads. Some insight into the structure and format of this program is provided.

## Introduction

Roads with posted speeds of 70km/h or higher introduce a considerable level of risk to workers responding to emergencies in these environments e.g. [1–3]. This workforce includes emergency service workers such as law enforcement officers, firefighters, and emergency medical services as well as traffic management responders [1, 2]. Australian and international statistics indicate that workers operating on these roads are at significant risk of being injured or killed [4–6]. To illustrate, Australian statistics for 2016 identified that the fatality rate for incident responders (2.1 per 100,000 workers) was 1.5 times higher than the national rate while the serious injury claim rate was four times higher compared to all occupations (37.9 claims per 1,000

**Funding:** Funding for this study was provided by the Department of Education and Training, Workforce Training Innovation Fund.

**Competing interests:** No authors have competing interests.

employers) [6]. These statistics highlight the need for prevention activities that optimise the safety of these frontline workers.

There is currently limited research to guide the development of evidence-based prevention activities for incident responders working on high-speed roads. Training designed to improve safety has primary been driven through jurisdiction-based legislation and regulatory practice, with other training focused on unanticipated contingency planning [7] using group-based gaming simulations e.g. [8]. Although this approach to prevention is critical, current training does not well align with the characteristics of the working environment. Incident response on high-speed roads can be characterised by a high degree of uncertainty and interdependence. Uncertainty refers to the extent to which valued work roles are formalized [9] and occurs when outcomes are achieved through adapting to and initiating change, rather than complying with requirements of the work role [10]. In uncertain contexts, behaviour is less predictable as individuals adapt to the changing demands and conditions.

Interdependence refers to a system where outcomes are achieved through shared goals [11] rather than isolated to the responsibilities of individual workers [10]. Within the context of incident response on high-speed roads, jurisdictional legislation mandates roles and responsibilities for each responding agency; however, often at times work tasks need to be shared to achieve a safe working environment. Training needs to align with the working context to optimise its effectiveness.

Newnam and Oxley [12] argue that training in environments characterised by a high degree of uncertainty and interdependence (like high-speed roads) needs to target the development of cognitive and perceptual skills required for workers to adapt behaviour to respond to unplanned and safety critical events. That is, a safe working environment requires not only compliance with legislated practice but workers' engagement in proactive safety management [13]. The goal of this study is to explore factors that impact on safety outcomes (i.e., near misses, secondary incidents) for incident responders working on high-speed roads. The focus is on the relationship with (i) a range of safety behaviours (i.e., compliance, participatory, voice) (ii) willingness to engage in safety behaviours (i.e., safety motivation) and (iii) clarity in role-behaviour expectancies (i.e., role conflict). The results will provide a foundation for understanding the cognitive, behavioural and perceptual mechanisms facilitating safe working practices for this highly valued workforce. This information will be used to develop a program to promote the safety of incident responders with group activities that replicate the mindsets and behaviours required while attending incidents on high speed roads.

## Background

High-speed environments have been described as complex and dynamic for incident responders, partly due to the nature of the response (e.g., number of vehicles involved) and the location of the incident (e.g., built-up environment) [3]. To control risk in this environment, workers receive comprehensive safety-focussed training to ensure core safety activities are carried out to maintain a safe workplace. To illustrate, in the Australian State of Victoria, the lead agency for emergency management [i.e., Emergency Management Victoria] has developed procedures to guide the strategic and operational management of incident response across all agencies. Standard Operating Procedures (SOPs) define the core activities of each responding agency and cover behaviours like dynamic risk assessments and contingency planning see, [1]. These types of behaviours have been referred to as *safety compliance* in the research literature [14]. Compliance with SOPs is considered as a critical safety behaviour in the context of incident response; thus, it is expected that compliance with SOPs in the context of incident responders operating on high-speed roads facilitates a safe working environment.

Whilst SOPs are critical in creating a safe working environment, it must be acknowledged that emergencies are unplanned, unpredictable, and dynamic [1]. The inherent risks associated with the growing number of interactions between multiple elements in the road transport system have created a work environment unparallel to other workplace contexts [15]. As a consequence, workers in these environments often need to manage situations that escalate at a rapid speed [7]. There is no SOP to cover all possible types of situations. Therefore, workers often require close interactions and coordination within and across response agencies to effectively manage these safety critical events [1, 7]. That is, workers must engage in safety behaviours that go beyond their core activities. These types of behaviours are referred to as *safety participation* and have been defined as those actions that help to develop an environment that supports safety [14]. Research has found that both safety compliance and safety participation are critical in reducing workplace injuries [16]. However, Probst and Brubaker [17] found that safety motivation has a longer lasting effect on safety outcomes, with a six-month lagged effect on safety compliance. Thus, it is expected that participation in safety activities and motivation to do so, facilitates a safe working environment for incident responders operating on high-speed roads.

Beyond compliance and participation in creating a safe working environment, research has also focused on safety behaviours that extend further than their formal task description. These are referred to as safety citizenship behaviours [18]. Employee safety voice is one of these [19, 20] and has been defined as communication motivated by the desire to achieve a safe working environment [20]. The role of employee safety voice in facilitating the exchange of information has received considerable research attention e.g., [13, 20, 21]. Research has shown that safety communication between workers and their managers e.g., [22–24] and peers/co-workers e.g., [20, 25] has a strong influence on safety behaviours. High levels of safety voice are associated with fewer work-related injuries when workers reported having a supervisor that was open to listening to their safety concerns [26]. Co-workers have also been found to play a critical role in encouraging individuals to speak up about safety issues [20]. This research suggests that employee safety voice is important in creating a safe working environment. Thus, it is expected that high levels of employee safety voice amongst incident responders is associated with less safety critical incidents.

Willingness to exert effort to create a safe working environment is a critical element of any safety system. Safety motivation has been defined as the willingness to put in effort to act safely and the importance an individual places on safe behaviours [14, 16]. Campbell, McCloy, Oppler, and Sager [27] identified motivation as a critical determinant of safety outcomes based on the understanding that behaviour is influenced by the motivational properties of the situation. In the context of incident responders, the motivation properties of mitigating risk to saving lives is likely to greatly determine the direction, amplitude and duration of actions taken by incident responders. In support, much research has shown a positive association between workers' motivation to participate in a safe working environment and safety outcomes [14, 16, 28, 29]. Thus, it is expected that motivation to engage in safe working practices on high-speed roads is associated with fewer safety incidents.

Engaging in safety behaviours and exerting effort to create a safe working environment are only two critical elements of a safe working environment; there also needs to be clarity in role-behaviour expectancies for incident responders working in these environments. Newnam et al. [1] showed that poor communication between incident response agencies can prevent the effective coordination of safety efforts between agencies and that such situations are often perpetuated by a lack of clarity in role-behaviour expectancies [1]. Although there are specific tasks that align with the core responsibilities of the agencies operating on high-speed roads (e.g., a paramedic's role is treating the injured person/s), some activities can be shared. For

example, the primary responsibility of a fire brigade is hazard mitigation and protecting the scene of the incident; however, fire brigades are also given the mandate to manage traffic if police are not at the scene of the incident [1]. This example illustrates that a lack of congruent expectations between and within job roles is likely to emerge, particularly in dynamic and unpredictable working environments.

The degree to which workers perceive predictability or clarity in their work tasks has been identified as an important factor in some frontline worker populations. For example, the degree to which nurses understand role-behaviour expectancies and expected performance goals is associated with negative safety outcomes [30]. Likewise, increased role clarity is associated with lower levels of stress in nurses [31]. This research highlights the importance of clarity in work tasks in safety critical contexts. Thus, it is expected that higher clarity in the work tasks when operating on high speed roads would be associated with a safer work environment.

## Research question

In summary, the aim of the study reported in this paper was to explore the behavioural (compliance, participation, voice), motivational (safety motivation) and work demand (role clarity) factors that influence safety outcomes for incident responders working on high-speed roads. The safety outcomes of interest in this study include near misses and secondary crashes. A near miss is defined as an unplanned event that does not result in injury or harm. A secondary incident is defined as an incident that occurred as a result of an original incident (for which the worker was responding). On high speed roads, secondary crashes are a known phenomenon. Tedesco and colleagues [32] reported that the risk of having another crash in the presence of an earlier crash can be six times higher. Further, for each additional minute the initial crash is present on a high-speed road, the likelihood of a secondary crash increases by 2.8% [33]. Thus, in the present study, participants were asked about the average number of secondary crashes per month they experience whilst on scene. This, along with near misses, gives a good indication of critical safety outcomes that can be influenced by individuals on scene.

Fig 1 describes the relationships explored in this research. This research will expand current knowledge on the factors that facilitate a safe working environment in dynamic and unpredictable work environments. The unique contribution of this research is in understanding the cognitive, behavioural and perceptual mechanisms facilitating safe working practices in a workforce characterised by a high degree of uncertainty and interdependence [10]. The information gained from this research will be used to inform the development of prevention activities to support existing SOPs with the goal of optimising communication and coordination practices in this highly valued workforce.

## Methods

### Participants and procedure

A total of 295 complete responses were received from three emergency service and two incident response agencies in one Australian State. Table 1 shows the sample demographics across agency. The participating agencies included the police, ambulance service, metropolitan fire brigade, and a private and public traffic management agency. Participants ranged in age from 21 to 66 years (M = 44.69; ±11.24) and on average had been in their current role for 11.47 years (± 9.85; range 0–43). Most of the sample (80%) were male.

This research was awarded ethical clearance from the host university. The survey was hosted in Qualtrics and recruitment was conducted via emails sent through management within the participating agencies. Completion of the survey was voluntary and consent was given based on completion of the survey (i.e., implied consent). Participant responses were

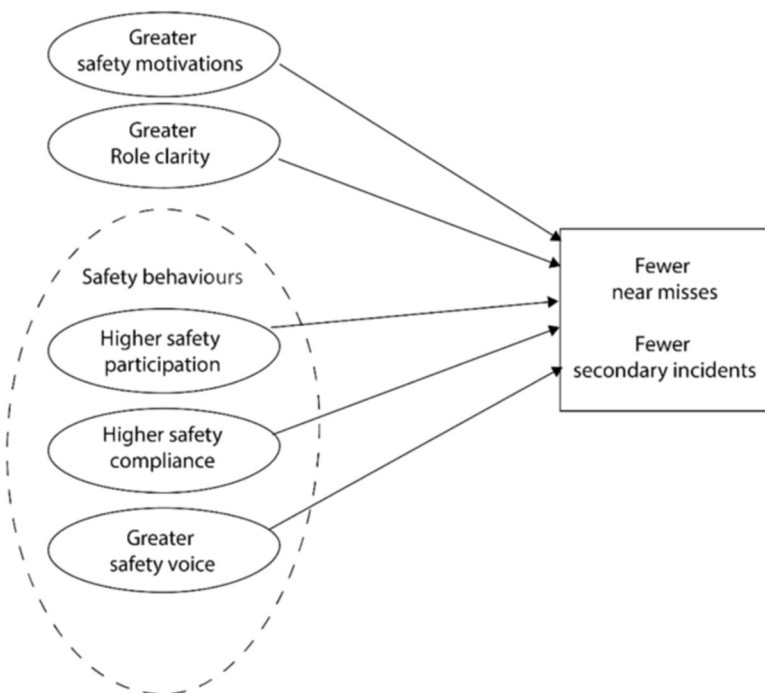

**Fig 1. Hypothesised relationships between safety behaviours, motivation, role clarity and safety outcomes.**

anonymous. Inclusion criteria required that workers attend incidents on high-speed roads as part of their work role.

## Measures

*Safety motivation* was assessed using a three-item scale developed by Neal and Griffin [16]. This scale measured participants' motivation to engage in safe working practices at the scene of an incident on high speed roads. The items were reworded to suit the working context. An example item is "I promote safety when attending an incident on a high-speed road". These items were measured on a 5-point Likert scale, ranging from strongly agree (1) to strongly disagree (5). Therefore, higher scores represent less safety motivation. Motivation has shown good reliability with Cronbach alphas of .85 and .92 measured on separate occasions [16].

*Safety compliance* was assessed using a three-item scale developed by Neal and Griffin [16]. This scale describes core safety activities that need to be carried out by individuals to maintain workplace safety (e.g., wearing PPE). The items were reworded to suit the working context. An example item is "I use all the required safety equipment to do my job when attending an

**Table 1. Age, gender and tenure among the five agencies (N = 295).**

| Agency | % total sample | Age (M:SD) | Gender: M/F | Tenure (M; SD) |
|---|---|---|---|---|
| Traffic management (private) | 3.7 | 48.5 (7.4) | 100/0.0 | 9.27 (3.1) |
| Police | 21.4 | 48.3 (9.0) | 88.9/11.1 | 10.90 (9.6) |
| Fire brigade | 24.7 | 45.9 (10.0) | 98.6/1.4 | 13.62 (11.4) |
| Ambulance[+] | 45.8 | 41.3 (12.3) | 62.2/37.8 | 10.87 (9.6) |
| Traffic management (public) | 4.4 | 51.1 (9.3) | 92.3/7.7 | 9.27 (3.1) |

[+] data do not equal 100 as one participant preferred not to say gender

incident on a high-speed road". These items were measured on a 5-point Likert scale, ranging from strongly agree (1) to strongly disagree (5), with higher scores indicated less safety compliance. This scale has shown good reliability with Cronbach alphas of .92 to .93 measured on separate occasions [16].

*Safety participation* was assessed using a three-item scale developed by Neal and Griffin [16]. This scale describes behaviours that may not directly contribute to workplace safety, but that do help to develop an environment that supports safety (e.g., voluntary safety activities). The items were reworded to suit the working context. An example item is "I promote safety when attending an incident on a high-speed road". These items were measured on a 5-point Likert scale, ranging from strongly agree (1) to strongly disagree (5). Therefore, higher scores on this scare represent less safety participation. This scale has shown good reliability with Cronbach alphas of .86 and .89 measured on separate occasions [16].

*Safety voice* was assessed using a four-item scale developed by Hofmann et al., [18]. This scale describes *extra-role* behaviours that actively promote safety in the work environment (e.g., speaking up about safety concerns). The items were reworded to suit the working context. An example item is "When attending the scene of an incident on a high-speed road, how often do you make safety-related recommendations?". These items were measured on a 5-point Likert scale, ranging from never (1) to frequently (5) with higher score therefore indicating more frequent extra role behaviours. This scale has shown good reliability with Cronbach alphas of .78 [13].

*Role clarity* was assessed using a four-item scale developed by Rizzo et al., [34]. This scale measures perception of clarity in respective roles and responsibility. The items were reworded to suit the working context. An example item is "When attending the scene of an incident on a high-speed road, how often do you have tasks you believe should be done in a different way?". These items were measured on a 5-point Likert scale, ranging from frequently (1) to never (5). Lower scores, therefore represented lack of clarity in role-behaviour expectancies. This scale has demonstrated good reliability with Cronbach alpha of .81 [34].

**Safety outcomes.**   Near misses and secondary incidents were measured by asking participants to state how often per month these incidents occur on high speed roads they attend.

**Control measures.**   Job tenure and exposure to incidents on high speed roads were included as control variables in this study. Tenure was measured by asking participants how long (years, months) they have worked in their current role. Exposure was measured by asking participants how often per month they attend incidents on high-speed roads.

## Data analysis and handling

Descriptive analyses were conducted in SPSS for Windows v.26. Individual relationships among variables were examined using Pearson correlations with strength cut-offs of < .20 for weak relationships, .20 to .40 for moderate relationships and r values above .40, deemed to represent strong relationships [35]. Simultaneous relationships between driver safety motivation, behaviours (compliance, participation, and voice), role clarity and safety outcomes (near miss and secondary incidents) were examined using Structural Equation Modelling (SEM) in AMOS for Windows v26. Tenure and average monthly incidents were controlled for in the model. A measurement model was initially fitted. For the initial full model (including the exogenous variables), all endogenous variables were covaried. However, non-significant covariances were freed from the final model.

The SEM used Maximum Likelihood estimates. A number of fit statistics were used to determine model fit as traditionally with larger samples, $X^2$ values are significant [36]. However, significant p values can be a sign of poor fit. Therefore, based on Hu and Bentler [37] and

Byrne [36] goodness of fit was determined by a Comparative Fit Index (CFI) of .90 or larger, Root Mean Square of Approximation (RMSEA) of .06 or less, a 90% Confidence Interval (CI) around the RMSEA with the upper value not exceeding .06 and a non-significant p-close (pclose > .05). As multivariate normality was not met (Kurtosis value >5), bootstrapping techniques on 2000 samples using the Bollen-Stine adjusted p value were used [38].

## Results

### Descriptive statistics

As can be seen in Table 2, participants reported attending an average of 7 incidents (M = 6.86; ±5.89) per month; experiencing 1 secondary incident (M = 1.32±2.19) per month and 3 (M = 3.49; ±3.97) near misses per month. Levels of safety motivation, compliance and participation were high, with average scores ranging between 1.32 to 1.51 (out of five) for these factors. Voice and role clarity were all frequent ranging from 3.75 to 4.13 (out of five). The safety variables all showed good reliability with Cronbach alphas ranging from .69 to .98.

Weak to moderate relationships were found between safety behaviours and safety outcomes of secondary incidents and near misses (rs ranging between -.12 and .31). Participation and role clarity were negatively related to average number of secondary incidents and near misses, indicating that participants who engaged in more activities to support safety in high speed environments also tended to be involved in fewer secondary incidents and near misses. Likewise, participants reporting lack of clarity in role-behaviour expectancies also tended to report higher average secondary incidents and near misses.

### Hypothesised relationships

The relationships between safety variables and outcomes were modelled with a SEM. The model showed good fit to the data $\chi^2_{(172)}$ = 280.24, p < .001; Bollen-Stine p =. 012; CFI = .98; RMSEA = .05 (90%CI: .04-.06), pclose = .73 and explained 29% and 28% of the variance in secondary incidents and near misses, respectively. As can be seen in Fig 2, after controlling for incidents per month and tenure, voice and role clarity were significant predictors of near

**Table 2. Intercorrelations among variables (with Means and SD).**

| | Age | Years in role | Incidents/ month | Secondary incidents/month | Near Misses/ month | Motivation | Compliance | Participation | Voice | Role clarity |
|---|---|---|---|---|---|---|---|---|---|---|
| Age | – | .59*** | .24*** | .05 | .10 | -.05 | -.11 | -.20*** | .09 | -.04 |
| Years in role | | – | .01 | -.07 | -.04 | .03 | -.07 | -.07 | -.03 | -.11 |
| Incidents/month | | | – | .47*** | .45*** | -.09 | -.04 | .12* | .10 | -.11 |
| Secondary incidents/month | | | | – | .56*** | .05 | .02 | -.12* | .19*** | -.19*** |
| Near miss/month | | | | | – | -.08 | -.10 | -.20*** | .31*** | -.19*** |
| Motivation | | | | | | – | .67*** | .51*** | -.10 | .03 |
| Compliance | | | | | | | – | .68*** | -.17*** | -.05 |
| Participation | | | | | | | | – | -.41*** | -.02 |
| Voice | | | | | | | | | – | .07 |
| Role Clarity | | | | | | | | | | – |
| α | na | na | na | na | na | .98 | .95 | .88 | .90 | .69 |
| Means (SD) | 44.70 (11.24) | 11.47 (9.85) | 6.86 (5.89) | 1.32 (2.19) | 3.49 (3.97) | 1.32 (.94) | 1.46 (.87) | 1.51 (.76) | 3.75 (.93) | 3.56 (.70) |

***p≤.001; ***p≤.05; α = Cronbach alpha

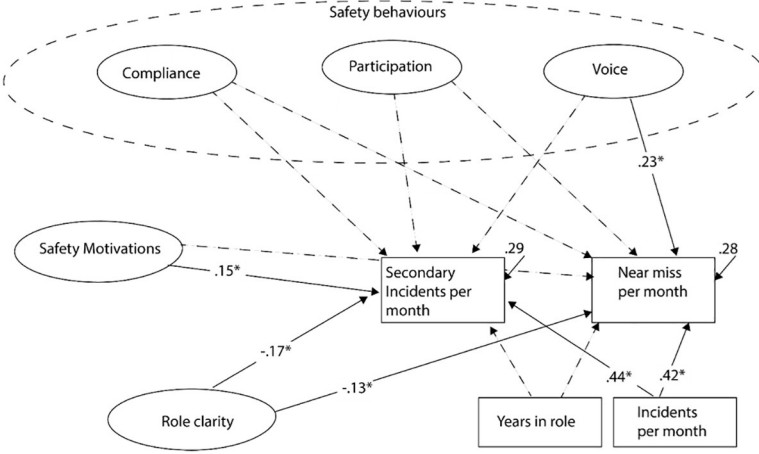

,*p <.01; dashed line = non significant;
X2 (172) = 280.24, p <.001; Bollen-Stine p = .012; CFI = .98,  RMSEA = .05, 90% CI = .04-.06, pclose = .73

**Fig 2.  Relationships between safety behaviours, motivations, role clarity and safety outcomes.**

misses, while role clarity and safety motivations were significant predictors of secondary incidents per month. Years in the role was unrelated to safety outcomes, while incidents per month predicted both secondary incidents and near misses. As expected, as the number of incidents attended increased, so did the number of secondary incidents and near misses. These findings partly support the hypotheses as no significant relationships were identified for safety compliance, safety participation and the safety outcomes.

Table 3 shows the relationships between the exogenous variables that were retained in the model. Non-significant relationships were freed from covarying in the model. Years in role was unrelated to the other exogenous variables and role clarity only weakly, negatively, related to incidents per month. Disturbance terms for endogenous variables were covaried and had weak relationships (r = .39, p < .001).

## Discussion

The goal of this research was to explore the behavioural (compliance, participation, voice), motivational (safety motivation) and work demand (role clarity) factors that influence safety outcomes for incident responders working on high-speed roads. This study presents a significant contribution to existing knowledge as limited research has explored safety in work environments characterised as uncertain and interdependent see, [23, 29]. This research is

**Table 3.  Relationships between exogenous variables.**

|  | Compliance | Participation | Voice | Role clarity |
|---|---|---|---|---|
| Safety motivations | .70*** | .55*** | na | na |
| Compliance | – | .76*** | -.12* | na |
| Partipation | na | – | -.39*** | na |
| Incidents per month | na | na | na | -.13* |

Na = not significant and removed

*** p ≤.001; **p ≤.01

*≤.05

important as safety can only be optimised if training aligns with the context of the organisation. The findings from this study will be used to guide the development of a feasible and practicable program to improve the cognitive, behavioural and perceptual skills of incident responders operating on high-speed roads.

Consistent with previous research, the results of this study identified that safety voice e.g., [20, 22–25] was a significant predictor of near misses, but not secondary incidents. That is, safety communication between workers (including managers and co-workers) at the scene of an incident is important in reducing the occurrence of near misses. Therefore, encouraging workers to speak up about safety issues or concerns at the scene of an incident is likely mitigate the risk of injury or harm at the scene of an incident. The lack of observed relationship with secondary crashes may reflect, in part, that some secondary crashes are less under individual control and may have more to do with the incident location or volume of passing traffic.

Motivation to engage in safe working practices was also positively associated with secondary incidents. This finding supports previous research [14, 16, 28, 29] by showing that willingness to exert effort in safety activities is associated with positive safety outcomes. In the context of this study focussing on incident responders, these findings highlight the importance of safety motivation. For example, the motivational properties of the work tasks can influence the propensity of incident responders to engage in behaviours to actively avoid an incident from occurring as a result of the original incident.

This study also found a relationship between role clarity and both safety outcomes. This result supports the findings of Newnam et al. [1] is so far as lack of clarity in role-behaviour expectancies was identified as a factor preventing the effective coordination of safety efforts between incident response agencies when operating on high-speed roads. This finding was not surprising given that the landscape of role-behaviour expectancies in safety management for incident responders on high-speed roads is complicated and changeable. Although there are clear jurisdictional guidance on SOPs, these guidelines do not cover all the contingencies that emerge in unpredictable and dynamic environments of high-speed roads. As a result, some activities relating to safety management can be shared across agencies, particularly when resources are limited at the scene of an incident or there is escalation in the incident. The results of this study suggest that a lack of congruent expectations between agencies and within job roles has the potential to create situations that increase the occurrence of near misses and secondary incidents.

It was somewhat surprising that no significant relationships were identified for safety compliance, safety participation and the safety outcomes considering the evidence of past research [14, 16, 17]. There is a clear requirement for incident responders to abide by the SOPs mandated by their agencies; however, the findings of this study suggest that compliance with SOPs is the only prevention approach to mitigate near misses and secondary incidents. This same argument applies for participation in activities to actively promote safety at the scene of an incident. It is apparent from these results that safety for incident responders on high-speed roads is only optimised through workers actively engaging in behaviours that go beyond what is expected and suggested within their formal role requirements.

## Practical implications

This study provides recommendations for improving the safety of incident responders operating on high-speed roads. Consistent with the findings of Newnam et al. [1], providing clarity in the roles and responsibilities of incident responders operating on high-speed roads has the potential to improve safety outcomes. Current interventions address this issue to some degree using gaming simulations to support group decision making focused on contingency planning

in emergency response see [8]. An alternative approach could be engaging incident responders in a group-based format to discuss ways of optimising communication and coordination in situations that require workers to think beyond the roles and responsibilities as mandated in their agencies' SOPs (i.e., escalated scenarios).

This approach to intervention not only has the potential to allow incident responders from different agencies to reflect on other workers' roles and responsibilities in safety-critical events, but it also has the potential to develop the communication skills of incident responders in voicing safety concerns and encouraging workers' willingness to engage in safety-related activities to promote a safe working environment. This approach supports the results of this study as well as previous research focused on optimising safety management skills in uncertain and interdependent contexts [12, 39].

The authorship team is currently designing a program to achieve these goals. This program draws upon the learnings of applied research see, [40], and identifies the importance of the key skills, communication and collaboration, in learning settings that utilise social learning and collaborative learning practices. The program is facilitator-led with group activities designed to replicate the mindsets and behaviours required while attending incidents on high speed roads. The goal of the activities is to promote conversation so to gain insight into the experiences, perspectives, and observations from each participant. Enhancing collaboration and communication within the group setting will not only enable the collective knowledge and understanding of the group to expand, but it will enable self-reflection and the acquisition of deeper insights at the individual level. The end-goal of this program is that the acquired knowledge and raised self-awareness will be taken back to the work environment in the form of increased collaboration and more effective safety communication. The results from the development and piloting of this program is forthcoming.

## Limitations

The limitations of this study need to be considered in the interpretation of findings. First, this research relied on self-report outcome measures which are open to socially desirable responding. However, the risk of bias is likely to be minimised as near misses and secondary incidents are less likely to affected by poor recall as they are recent (i.e. per month), salient and reportable events. Furthermore, accurate recall of workplace incidents has been found to be reliable in age groups 25–54 for up to a period of 12 months [41]. The average age of participants in this study was 44 years. Despite this support, objective safety outcomes will strengthen the accuracy of reporting.

Second, the use of cross-sectional design may have inflated the relationships identified in this study see [42]. For example, witnessing a near miss may result in a worker changing their work practices and voicing their safety concerns. Thus, caution should be taken in the inferences drawn from these results. Future research could overcome this issue through a longitudinal design.

Third, the results of this research may be representative of the legislature of the country in which this research was undertaken (Australia). Although the cognitive, behavioural and perceptual factors influencing safety in these working environments are likely to be the same across uncertain and interdependent working environments, the strength of these relationships may differ. Thus, the findings may have limited generalisability to other countries and states with different constitutions and different road infrastructures. Thus, recommendations to inform the development of evidence-based intervention should accurately capture the operational context to optimise effectiveness in achieving safety outcomes.

Finally, one outcome of interest in the present study was the occurrence of secondary crashes. This safety outcome may or may not be associated with the management of the

original scene (e.g., road infrastructure). This means that individuals on scene may not be able to control or influence the occurrence of all secondary crashes.

## Conclusion

This study presented an important investigation into the safety of incident responders operating on high-speed roads and identified that expectancies in work tasks, safety communication and motivation are critical in creating a safe working environment. The findings of this study not only extend current knowledge on factors that promote safety in uncertain and interdependent working contexts, they provide insight into the development of a program to improve the cognitive, behavioural and perceptual skills of incident responders operating on high-speed roads. The next stage in this program of research is to develop, implement and evaluate a program to improve the safety of incident responders operating on high-speed roads.

## Acknowledgments

The First Responder Safety on High Speed Roads project was supported by the Victorian Government.

## Author Contributions

**Conceptualization:** Sharon Newnam, Carlyn Muir, Tony Mazzeo.

**Data curation:** Amanda Stephens.

**Funding acquisition:** Sharon Newnam, Carlyn Muir, Tony Mazzeo.

**Investigation:** Sharon Newnam.

**Methodology:** Sharon Newnam, Carlyn Muir, Tony Mazzeo.

**Project administration:** Sharon Newnam.

**Writing – original draft:** Sharon Newnam, Amanda Stephens, Simon Bruce, Tim Austin.

**Writing – review & editing:** Sharon Newnam, Amanda Stephens, Carlyn Muir, Simon Bruce, Tim Austin, Tony Mazzeo.

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
