## [Decision Letter · Decision Letter 0]

2 Feb 2021

Safety outcomes for incident responders operating on high speed roads: An analysis of the relationship with behaviour, motivation and role clarity

PONE-D-20-37530

Dear Dr. Newnam,

We’re pleased to inform you that your manuscript has been judged scientifically suitable for publication and will be formally accepted for publication once it meets all outstanding technical requirements.

Kind regards,

Feng Chen

Academic Editor

PLOS ONE

1. Please provide additional details regarding participant consent. In the ethics statement in the Methods and online submission information, please ensure that you have specified what type you obtained (for instance, written or verbal, and if verbal, how it was documented and witnessed). If your study included minors, state whether you obtained consent from parents or guardians. If the need for consent was waived by the ethics committee, please include this information.

2.Thank you for stating the following in the Financial Disclosure section:

"Funding for this study was provided by the Department of Education and Training, Workforce Training Innovation Fund"

We note that one or more of the authors are employed by a commercial company: Broadspectrum, Transport ANZ and Visual Learning Design

Please respond by return email with an updated Funding Statement and Competing Interests Statement and we will change the online submission form on your behalf.

Reviewers' comments:

Reviewer's Responses to Questions

**Comments to the Author**

1. Is the manuscript technically sound, and do the data support the conclusions?

Reviewer #1: Yes

Reviewer #2: Yes

2. Has the statistical analysis been performed appropriately and rigorously? 

Reviewer #1: Yes

Reviewer #2: Yes

3. Have the authors made all data underlying the findings in their manuscript fully available?

Reviewer #1: Yes

Reviewer #2: Yes

4. Is the manuscript presented in an intelligible fashion and written in standard English?

Reviewer #1: Yes

Reviewer #2: Yes

5. Review Comments to the Author

Reviewer #1: This paper presents an analysis of the effects of factors specific to behavior, motivation and role clarity on safety outcomes for incident responders operating on high speed roads. The research topic is interesting and worth of investigation. The structural equation model proposed for the empirical analysis is reasonable. Overall, the paper is well written and organized. I recommend its publication in PLOS ONE.

Reviewer #2: The current manuscript examines the behavioral, motivational, and work demand factors that impact the safety outcomes of incident workers on high speed road. The topic is interesting and worthy of investigating. Overall, the paper is well written, the methodology is sound, the conclusion is well supported by the results.

6. PLOS authors have the option to publish the peer review history of their article (what does this mean?). If published, this will include your full peer review and any attached files.

Reviewer #1: No

Reviewer #2: No

---

## [Editor Report · Acceptance letter]

10 Feb 2021

PONE-D-20-37530 

Safety outcomes for incident responders operating on high speed roads: An analysis of the relationship with behaviour, motivation and role clarity 

Dear Dr. Newnam:

I'm pleased to inform you that your manuscript has been deemed suitable for publication in PLOS ONE. Congratulations! Your manuscript is now with our production department. 

Kind regards, 

on behalf of

Dr. Feng Chen 

Academic Editor

PLOS ONE